# Predicting Breast Cancer Risk Using Radiomics Features of Mammography Images

**DOI:** 10.3390/jpm13111528

**Published:** 2023-10-25

**Authors:** Yusuke Suzuki, Shouhei Hanaoka, Masahiko Tanabe, Takeharu Yoshikawa, Yasuyuki Seto

**Affiliations:** 1Department of Breast and Endocrine Surgery, Graduate School of Medicine, The University of Tokyo, 7-3-1 Hongo, Bunkyo-ku, Tokyo 113-8655, Japan; 2Department of Radiology, Graduate School of Medicine, The University of Tokyo, 7-3-1 Hongo, Bunkyo-ku, Tokyo 113-8655, Japan; hanaoka-tky@umin.ac.jp; 3Department of Computational Diagnostic Radiology and Preventive Medicine, The University of Tokyo Hospital, 7-3-1 Hongo, Bunkyo-ku, Tokyo 113-8655, Japan

**Keywords:** breast cancer, Light GBM, mammography, radiomics, risk prediction

## Abstract

Mammography images contain a lot of information about not only the mammary glands but also the skin, adipose tissue, and stroma, which may reflect the risk of developing breast cancer. We aimed to establish a method to predict breast cancer risk using radiomics features of mammography images and to enable further examinations and prophylactic treatment to reduce breast cancer mortality. We used mammography images of 4000 women with breast cancer and 1000 healthy women from the ‘starting point set’ of the OPTIMAM dataset, a public dataset. We trained a Light Gradient Boosting Machine using radiomics features extracted from mammography images of women with breast cancer (only the healthy side) and healthy women. This model was a binary classifier that could discriminate whether a given mammography image was of the contralateral side of women with breast cancer or not, and its performance was evaluated using five-fold cross-validation. The average area under the curve for five folds was 0.60122. Some radiomics features, such as ‘wavelet-H_glcm_Correlation’ and ‘wavelet-H_firstorder_Maximum’, showed distribution differences between the malignant and normal groups. Therefore, a single radiomics feature might reflect the breast cancer risk. The odds ratio of breast cancer incidence was 7.38 in women whose estimated malignancy probability was ≥0.95. Radiomics features from mammography images can help predict breast cancer risk.

## 1. Introduction

The importance of identifying women at high risk of developing breast cancer has increased in recent years. Globally, breast cancer affects more women than any other cancer [1]. Early detection and treatment can achieve a high cure rate. Mammography images contain a lot of information about the breasts (mammary gland, stroma, breast size, breast shape, skin properties, and intramammary fat). Using this information, mammography screening has contributed to the early detection of breast cancer and reduced mortality [2,3,4,5,6]. However, some tumors are difficult to detect as malignant tumors using mammography owing to the location of the breast tumor [7,8,9], tumor size [8], tumor density [8], and dense breasts [10,11]. The presence of these non-technical false-negative cases is a weakness of mammography-only screening.

Therefore, using other modalities, such as digital breast tomosynthesis [12], ultrasonography [13], computed tomography (CT), and magnetic resonance imaging (MRI) [14], is essential. However, the application of these modalities needs to be limited to high-risk cases [15], such as those with *BRCA1/2* pathogenic variants [16]. This is because they may increase the false positive rate and radiation exposure. Furthermore, they have not been shown to reduce mortality [14,17].

Various models have been devised to calculate the risk of developing breast cancer. The most well-known is the Gail model [18]. However, its accuracy is not very high because it calculates the risk from clinical information alone. In recent years, the addition of genetic information, such as single-nucleotide polymorphisms, is expected to improve accuracy [19,20]. However, it is not realistic to utilize genetic testing for everyone to incorporate genetic information because of the high cost, disadvantages associated with the test results (discrimination and anxiety before the onset of disease due to genetic information), and the fact that the test results may also affect blood relatives. Other risk calculation models have used mammography images [20,21]. In recent reports, risk assessment was performed using not only mammary gland density but also the texture of the mammary gland area [22].

Several changes are known to occur in the mammary glands of women at high risk for breast cancer. For example, diabetes, a risk factor for breast cancer development [23], can cause a characteristic change in the mammary glands of patients with diabetes, known as diabetic mastopathy. This change can be detected using ultrasonography and may also be identified on mammography images [24,25].

Furthermore, *BRCA1/2* pathogenic variants are reflected in MRI images [26]. In this method, radiomics features of MRI images are utilized and considered useful for extracting genetic information from medical images. Radiomics can extract texture and shape features from radiological images that are invisible to the naked eye and is one of the most useful methods for analyzing specific regions in radiological images [27].

Radiomics has been used in various modalities in the field of breast cancer [28]. In recent years, a number of studies, particularly those on mammography [29], have reported its usefulness not only for predicting whether a tumor is benign or malignant [30,31], but also its molecular subtypes [32,33], risk of recurrence [34], and prognosis [35]. Thus, a lot of information can be obtained from the radiomics features of mammography images; therefore, the risk of developing breast cancer can also be calculated, leading to individually optimized breast cancer screening methods [36].

In fact, some pilot studies have demonstrated the potential for predicting the risk of breast cancer development from radiomics features of mammography using small numbers and closed datasets [22,37]. In particular, Zheng et al. [37] reported that the risk of breast cancer development could be predicted with very high accuracy (area under the curve (AUC): 0.85) using radiomics features of mammography. However, further validation using large-scale or public datasets remains to be reported.

In this cross-sectional study, we used a large and publicly accessible dataset. Radiomics features calculated from mammography images of a healthy-side breast in women with breast cancer were compared with those of breast cancer-free women to estimate the individual risk of developing breast cancer and to identify radiomics features associated with the risk of developing breast cancer.

## 2. Materials and Methods

Firstly, we hypothesized that bilateral breasts in a woman have the same environment, such as a germline pathogenic variant [38], exposure to estrogen [39], obesity [40], and lifestyle [41], which would lead to breast cancer development. In breast cancer treatment, some reports suggest that the outcome can be predicted from the image data of the healthy-side breast [42]. Thus, it is assumed that images obtained from the contralateral breast of a patient with breast cancer reflect information about the whole-body environment.

Based on this hypothesis, we compared radiomics features of mammography images of patients with breast cancer and breast-cancer-free women to predict the risk of developing breast cancer. For the malignant group, we used the mammography images from the healthy side, whereas for the breast-cancer-free women, we randomly selected mammography images from either breast to match the left–right ratio of the patients with breast cancer during validation. Using radiomics features calculated from whole-breast mammography images in both groups, we could predict the breast cancer risk and identify specific radiomics features associated with an increased risk of breast cancer development.

### 2.1. Dataset

We used the OPTIMAM Mammography Image Database [43] as our mammography dataset. This dataset contains screening mammography images and patient data collected from three institutions in the United Kingdom since 2011: the Jarvis Breast Screening Centre in Guildford, St George’s Hospital in southwest London, and Addenbrooke’s Hospital in Cambridge. The entire dataset consisted of 154,832 normal, 6909 benign and 9690 malignant cases and 1888 intermediate cancers in 173,319 women; however, we received 1000 normal, 1000 benign, and 4000 malignant cases as the ‘starting point set’. The 4000 malignant cases were defined as the malignant group and the 1000 normal cases as the normal group. However, the 1000 benign cases were not used.

For the malignant group, we used mammography images obtained just before biopsy to diagnose breast cancer. For the normal group, we used cases that were judged as normal in mammography images more than one year later. We used the oldest image for the normal group.

The proportion of mammography equipment manufacturers for the normal and malignant groups is shown in Table 1. The proportion of malignant cases was higher in those taken with Philips equipment than with other manufacturers. To control for potential confounding effects due to differences in equipment, we only used images obtained with the most frequently used Hologic mammography device (85.1% in the malignant group and 92.0% in the normal group).

We also excluded cases with artificial objects in the images (implants, piercings, implanted devices for cardiac disease), large breast sizes that prevented routine imaging, bilateral breast cancer, and one-sided mammography images. A summary of the excluded cases is shown in Table 2. Finally, 3215 malignant cases and 896 normal cases were included in the analysis.

All 4111 cases that met the inclusion criteria had mammography images available in two directions, mediolateral oblique (MLO) and craniocaudal (CC). However, because of substantial inter-individual variability in MLO-view images arising from factors such as the breast compression angle, success of pectoral muscle compression, and presence of abdominal subcutaneous fat, we chose to use only CC-view images in our study. For the malignant group, we used CC images from the healthy breast. For the normal group, we randomly selected bilateral CC images to match the left–right ratio of the malignant group.

### 2.2. Breast Mask Image

We created a mask image of the breast region for each breast. For each image, a mask was created using a threshold value of 1/100, numerically calculated with Otsu’s binarization method [44]. In some cases, creating the mask failed with this method, but we were still able to create masks using the threshold value calculated using Otsu’s binarization method.

### 2.3. Radiomics Features

Radiomics feature values for all breast regions were calculated using Pyradiomics, an open-source Python package platform (http://www.radiomics.io/pyradiomics.html, accessed on 14 September 2023). The radiomics features used in this study were classified into seven categories: shape-based 2D, First Order Statistics, Gray Level Cooccurrence Matrix (GLCM), Gray Level Run Length Matrix (GLRLM), Gray Level Size Zone Matrix (GLSZM), Gray Level Dependence Matrix (GLDM), and Neighborhood Gray Tone Difference Matrix (NGTDM). Shape-based 2D features were calculated only from the original mammography images. The other six categories of radiomics features were calculated from both the original mammography images and the mammography images processed with six filters (log (laplacian of gaussian)-sigma-2-0 mm, log-sigma-3-0 mm, log-sigma-4-0 mm, log-sigma-5-0 mm, wavelet-H, wavelet-L). As a result, a total of 646 radiomics features were obtained. The feature items used in this study (https://pyradiomics.readthedocs.io/en/latest/features.html, accessed on 14 September 2023) are shown in Figure 1.

### 2.4. Analysis

All machine learning analyses and statistical processing were performed in Python 3.6.5.

We used the Light Gradient Boosting Machine (LGBM) (https://lightgbm.readthedocs.io/en/v3.3.2/) for our classification task. LGBM is an open-source distributed gradient-boosting framework developed by Microsoft Corporation that uses supervised learning to compute the objective variable from the explanatory variables using a decision tree method. Unlike the sort-based decision tree algorithm used in eXtreme Gradient Boosting and other implementations, LGBM features a highly optimized histogram-based decision tree learning algorithm, which increases efficiency and reduces memory consumption.

We conducted the performance evaluation of the classification task using a k-fold cross-validation scheme. The cases were randomly divided into k equal subsets, and the evaluation was performed k times, each time using a different subset as the test set. In this study, k was set to 5 to ensure a sufficient number of normal and malignant cases in each subset. The k-fold cross-validation scheme is effective in preventing models from overfitting.

We trained and tested the LGBM with the target variable being whether the case was malignant or normal and the explanatory variable being radiomics features. We calculated the accuracy and AUC for each split. Due to the age bias in the dataset, we conducted analyses limited to all ages and individuals in their 50s or 60s to control for age effects. We verified whether the AUC for each age group significantly exceeded 0.5 using a one-sample *t*-test.

The probability of malignancy for each case was predicted using the trained LGBM. The threshold for the malignant probability for calculating the odds ratio of being the contralateral breast of a malignant tumor was set at 0.95.

In the classification task distinguishing malignant from normal cases, we identified the radiomics features that were frequently involved in the decision tree branching. To verify the distribution differences of these radiomics features between malignant and normal cases, we conducted a covariance analysis adjusted for age.

## 3. Results

The cases used in the analysis, categorized by age and by left and right sides of the mammographic image, are shown in Table 3. All LGBM parameters were set as default values, except n_estimators = 1,000,000, learning rate = 0.0001, and class_weight = ‘balanced’. The AUC, accuracy, and receiver operating characteristic curves calculated from the five-fold cross-validation are shown in Table 4 and Figure 2a–c for each fold. The AUC exceeded 0.5 in all folds for all cases and groups in their 50s and 60s. However, the AUC was lower in those in their 50s and 60s than in all cases, possibly due to the small number of training cases. We performed a one-sample t-test for the AUC of each age group and confirmed that each AUC was significantly above 0.5 (all ages: *p* < 0.001, 50s: *p* = 0.030, 60s: *p* = 0.002).

We present histograms showing the probability of malignancy for each case in the malignant and normal groups in Figure 2d–e. The histograms of the malignant cases have a peak value further to the right than the histograms of the normal cases. Furthermore, the histograms of the malignant cases also have many counts (n = 103, 3.3%), wherein the probability of being malignant is >0.95 (red bar), whereas the normal cases have a very small number of counts (n = 4, 0.4%). The odds ratio of being a malignant case is 7.38 (i.e., (103/4)/(3112/892)), using a cut-off value of the probability of malignancy > 0.95.

The normal group has equal numbers of left and right sides, as both sides are available.

Some features were found to be important as they were frequently used in decision tree branching across multiple folds (Table 5). For example, ‘wavelet-H_glcm_Correlation’ and ‘wavelet-H_firstorder_Maximum’ ranked among the top 10 in terms of importance in all folds, and ‘original_glrlm_LongRunLowGrayLevelEmphasis’ ranked among the top 10 in terms of importance in four folds.

We present 10 radiomics features that were the top 10 most frequently used for decision tree branching across all folds and histograms of the values of these features for both normal and malignant groups (Figure 3). We analyzed the differences in the distribution in each radiomics feature in the normal and malignant groups after adjustment for age. Significant differences in the following three radiomics features were observed between the normal and malignant groups even after age adjustment: ‘wavelet-H_glcm_Correlation’ (*p* < 0.001), ‘wavelet-H_firstorder_Maximum’ (*p*< 0.001), and ‘wavelet-H_glcm_Imc2’ (*p* < 0.001) (Figure 3).

## 4. Discussion

In recent years, deep learning has been widely used in the field of image analysis with the advancement of computational equipment. Deep learning-based methods might also be able to predict the risk of developing breast cancer using mammography images of the contralateral breast. Nevertheless, for the classification tasks in this study, we chose to use LGBM, a decision tree method. There are two reasons for this. One is that there have been reports that machine learning is more useful than deep learning when the difference between the number of radiomics features used as explanatory variables and the number of cases is not sufficient [45]. In this study, we used a large number of radiomics features, so we decided to use the decision tree method. Another reason is that using decision tree methods allows us to provide explanations for classification using radiomics features. It is challenging to explain which elements of each image are used for classification in deep learning methods. In contrast, we were able to identify several radiomics features, such as ‘wavelet-H_glcm_Correlation’ and ‘wavelet-H_firstorder_Maximum’, which may be useful in predicting the risk of breast cancer development. It is desirable to verify whether these features are related to known breast cancer risk factors, such as genetic mutations [16], smoking [41], and long-term estrogen exposure [39], in the future.

While only the mammary gland area was used to calculate radiomics features in previous studies [22,37], the whole breast was used in our study. This has two advantages. One is that it is more versatile and simpler, as no specific breast region is extracted. The second is that information from outside the mammary area (skin properties and adipose tissue area) can also be incorporated into the model. Biologically, we know that parenchymal stromal cells and adipocytes in the breast influence the development and progression of breast cancer, and that estrogen receptors, which are largely responsible for breast cancer development, are also expressed on epidermal and dermal cells [46]; therefore, information from outside the mammary gland region may also reflect the risk of developing breast cancer.

However, this approach reflects mammary gland density, which has certainly been correlated with breast cancer development in previous reports [21]. Therefore, there was a concern that the results of this study might simply reflect mammary gland density. As a result, the features related to classifying the malignant and normal groups in this study were not strongly and directly correlated with mammary gland density, such as ‘firstorder_original_mean’.

Therefore, this suggests that some radiomics features that are not highly associated with mammary gland density are associated with the risk of developing breast cancer. The AUC for the risk of developing breast cancer estimated from mammary gland density alone in a previous study using 1 million mammography images with approximately 10,000 malignant cases was 0.57 [47], whereas the AUC obtained from whole-breast radiomics features in this study was equal to or greater than this value. This suggests that the whole-breast radiomics features reflect the risk of developing breast cancer, and ‘wavelet-H_glcm_Correlation’ and ‘wavelet-H_firstorder_Maximum’ are good candidates for this, as they are frequently used in decision tree branching and show statistically significant differences between the normal and malignant groups.

The relationship between these radiomics features and the risk of developing breast cancer can also be visually observed in histograms of relative frequency densities. For ‘wavelet-H_glcm_Correlation’, high bars for malignant cases can be seen to the left of the peak, whereas for ‘wavelet-H_firstorder_Maximum’, high bars for malignant cases can be seen to the right of the peak. The former feature, ‘wavelet-H_glcm_Correlation’, represents local similarity with neighboring pixels after high-frequency component emphasis (using wavelet transform) [48]. The latter feature, ‘wavelet-H_firstorder_Maximum’, represents the maximum pixel value with high-frequency component emphasis. Therefore, higher local inhomogeneity and maximal intensity in high-frequency emphasized images are related to the risk of developing breast cancer.

Some other radiomics features show differences in distribution between the malignant and normal cases in the relative frequency density histograms, while others do not show much difference. These features that do not show differences in the relative frequency density histograms were considered to be features that do not reflect the risk of developing breast cancer when used as stand-alone radiomics features. However, when combined with other radiomics features, they may contribute to risk estimation for developing breast cancer.

In this analysis, using all 646 radiomics features, we were able to calculate the probability of having any cancer in the contralateral breast. When we set the cut-off threshold of this probability value as 0.95, the odds ratio of having the malignancy in the contralateral breast was 7.38. Therefore, this value can be interpreted as the relative risk ratio of the high-risk group, which is defined in this study. As such, it may be possible to recommend additional imaging examinations and closer screening schedules for women whose probability of malignancy values exceed 0.95 but do not show abnormalities on mammography. A sufficiently high pre-test probability may justify following ultrasound and/or contrast-enhanced MRI as a further breast cancer screening.

In recent years, prophylactic treatment, medication, and surgical treatment have also been considered for individuals at high risk of developing breast cancer [49,50].

Identifying high-risk individuals using our method may aid in selecting appropriate prophylactic medication targets. The accuracy of this approach is expected to improve as more cases are accumulated. Furthermore, machine learning using many images generally requires significant data preparation and cleaning efforts, such as setting regions of interest. However, this method is simple and has the advantage that the model can be easily enhanced even after accumulating a large number of cases.

In this study, we did not determine what events in the breast tissue were reflected in the radiomics features suggested to be associated with breast cancer risk. These features may reflect changes in the mammary tissue or adipose tissue due to hormonal balance, blood glucose levels, or other factors such as genetic mutations. Further investigation using datasets linked to clinical data or genetic information is needed to clarify this point.

Our study’s limitations include the retrospective nature of the investigation and the use of images of the contralateral breast in women with breast cancer. To evaluate the results of this study, observational studies on the population-calculated radiomics features from mammography images are needed in the future. Another limitation is that the mammography images used were from a dataset and a single imaging equipment manufacturer. It is necessary to verify whether the results of this study apply to other datasets and mammography equipment from different manufacturers.

Finally, the AUC for breast cancer risk prediction obtained in this study is not very high. It is slightly higher than the AUC of the Gail model (AUC: 0.55 (0.52–0.57)) and the Tyrer–Cuzick model (AUC: 0.57 (0.55–0.59)) alone, and is comparable to the AUC obtained by adding mammary gland density information to the Gail (AUC: 0.59 (0.57–0.61)) and Tyrer–Cuzick models (AUC: 0.61 (0.59–0.63)) [51]. Therefore, improving accuracy with larger data sets, comparison with these known breast cancer risk models, and verification of synergistic effects are needed and will be the focus of our future work.

## 5. Conclusions

In conclusion, our study demonstrates that radiomics features obtained from mammography images using a simple method can predict the risk of developing breast cancer. We identified three radiomics features, ‘wavelet-H_glcm_Correlation’, ‘wavelet-H_firstorder_Maximum’, and ‘wavelet-H_glcm_Imc2’, which might reflect the risk of developing breast cancer. Furthermore, using our method, it is suggested that the subgroup with a 7.38-fold relative cancer prevalence risk can be identified. We hope that incorporating these radiomics features into breast cancer screening will lead to the addition of other examination modalities to reduce mortality rates and help identify individuals at high risk of developing breast cancer for prophylactic treatment.

## Figures and Tables

**Figure 1 jpm-13-01528-f001:**
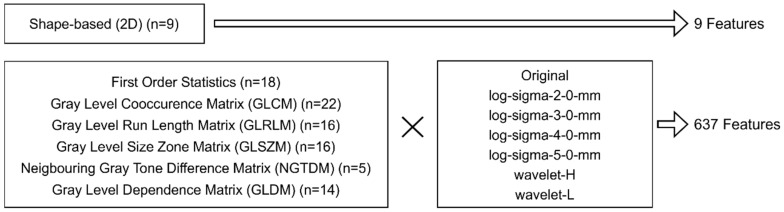
A summary of the radiomics features. Nine shape-based features were used as is, while the other features were calculated using six different image transformations in addition to the original image.

**Figure 2 jpm-13-01528-f002:**
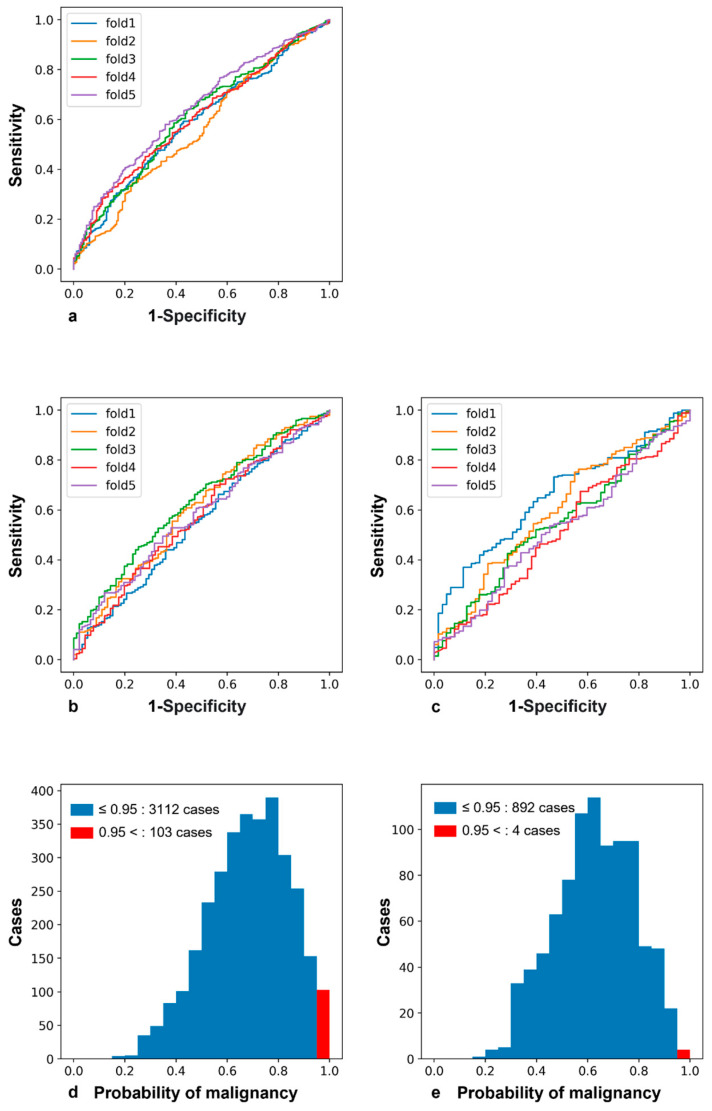
(**a**–**c**) Receiver operating characteristic curves for each fold at all ages (**a**), 50s (**b**), and 60s (**c**). Histograms of probabilities of malignancy for the malignant (**d**) and normal (**e**) groups, predicted from radiomics features.

**Figure 3 jpm-13-01528-f003:**
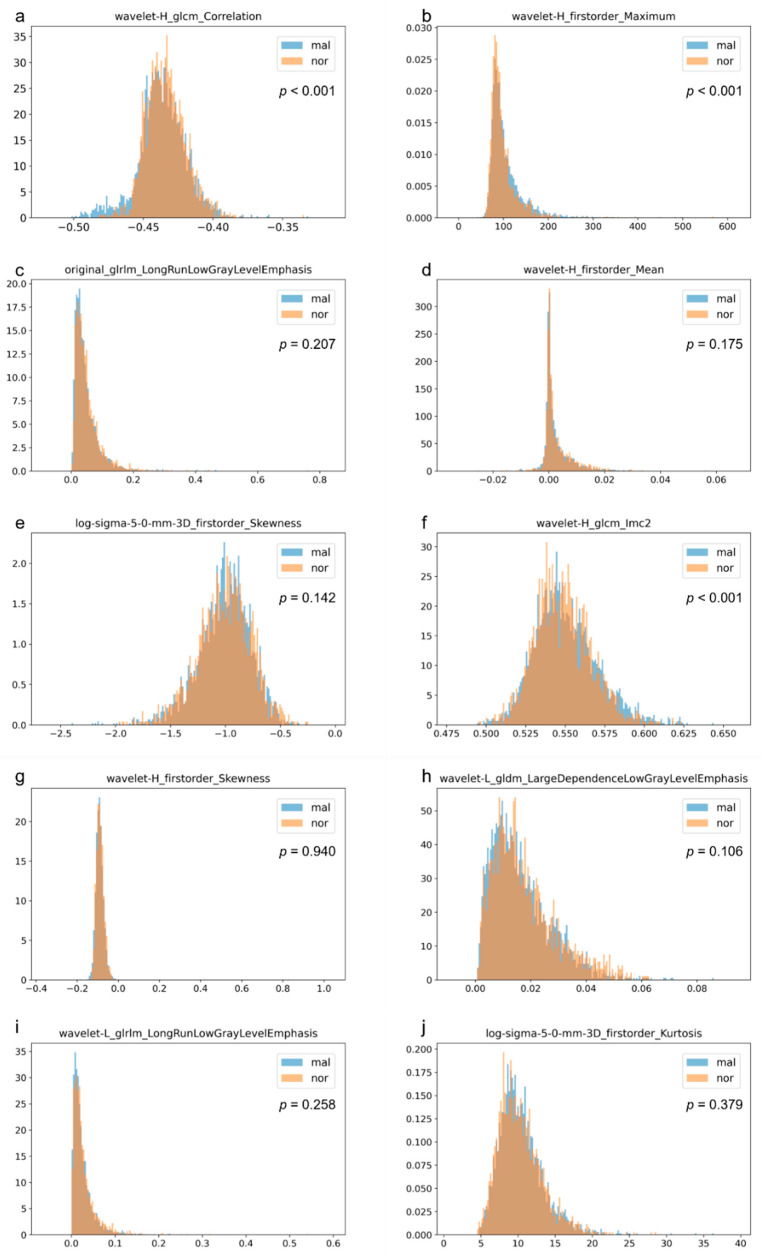
Histograms of relative frequency densities of the malignant and normal groups for each radiomics feature used most frequently for decision tree branching and significant differences (*p*-values) between the normal and malignant groups after age adjustment for each radiomics feature. Each radiomics feature was used more frequently in alphabetical order (**a**–**j**).

**Table 1 jpm-13-01528-t001:** Ratio of mammography equipment in the malignant and normal groups.

Manufacturer of Mammography	Malignant	Normal
Hologic	3403 (85.1%)	920 (92.0%)
GE	128 (3.2%)	43 (4.3%)
Philips	341 (8.5%)	13 (1.3%)
SIEMENS	126 (3.2%)	23 (2.3%)
Sectra Imtec	0 (0.0%)	1 (0.1%)
No image	2 (0.1%)	0 (0.0%)
All	4000	1000

**Table 2 jpm-13-01528-t002:** Reasons for exclusion of cases in the malignant and normal groups.

Reasons for Exclusion of Cases	
Malignant Case (n = 785)	Normal Case (n = 104)
Not Hologic (n = 595)	Not Hologic (n = 80)
Incidence of bilateral breast cancer (n = 123)	Breast implant (n = 12)
No healthy side images (n = 39)	Inadequate follow-up period (n = 7)
Breast implant (n = 13)	Image only of one side (n = 4)
Large breast (n = 7)	Foreign body reaction (n = 1)
Surgical history (n = 4)	
Artificial object (n = 2)	

**Table 3 jpm-13-01528-t003:** Distribution of participants based on age and left–right ratio.

	Malignant Group			Normal Group		
Age	Right	Left	All	Right	Left	All
<40	0	1	1	0	0	0
40≤, <50	117	110	227	101	101	202
50≤, <60	603	612	1215	457	457	914
60≤, <70	663	643	1306	312	312	624
70≤	240	227	467	26	26	52
All cases	1623	1592	3215	896	896	1792

**Table 4 jpm-13-01528-t004:** AUC and accuracy of each fold and their mean values in the 50s, 60s, and all cases.

	Area under the Curve			Accuracy		
Fold	50s	60s	All Cases	50s	60s	All Cases
fold 1	0.54701	0.65452	0.58894	0.64478	0.72840	0.71203
fold 2	0.60367	0.59813	0.55970	0.67365	0.75309	0.72384
fold 3	0.62954	0.55099	0.60930	0.68862	0.75309	0.72141
fold 4	0.56994	0.51286	0.60561	0.63174	0.68210	0.72384
fold 5	0.57981	0.52274	0.64254	0.64072	0.75232	0.72749
Average	0.58599	0.56785	0.60122	0.65590	0.73380	0.72172

**Table 5 jpm-13-01528-t005:** Top 10 most frequently used radiomics features for decision tree branching in each fold.

Radiomics Features Name		
	fold 1	fold 2
1	wavelet-H_firstorder_Mean	log-sigma-5-0-mm_firstorder_Skewness
2	wavelet-H_glcm_Correlation	wavelet-H_firstorder_Maximum
3	wavelet-H_firstorder_Maximum	original_glrlm_LongRunLowGrayLevelEmphasis
4	wavelet-H_glcm_Imc2	wavelet-H_glcm_Correlation
5	log-sigma-2-0-mm_firstorder_Mean	original_glcm_Idmn
6	wavelet-L_gldm_LargeDependenceLowGrayLevelEmphasis	wavelet-H_firstorder_Mean
7	log-sigma-5-0-mm_firstorder_Kurtosis	wavelet-H_glrlm_RunVariance
8	wavelet-H_ngtdm_Complexity	log-sigma-5-0-mm_firstorder_Kurtosis
9	wavelet-H_ngtdm_Contrast	log-sigma-4-0-mm_firstorder_Mean
10	log-sigma-3-0-mm_firstorder_Maximum	log-sigma-5-0-mm_firstorder_Maximum
	fold 3	fold 4
1	wavelet-H_firstorder_Maximum	wavelet-H_glcm_Correlation
2	wavelet-H_glcm_Correlation	wavelet-H_firstorder_Mean
3	wavelet-H_firstorder_Skewness	original_glrlm_LongRunLowGrayLevelEmphasis
4	original_glrlm_LongRunLowGrayLevelEmphasis	log-sigma-5-0-mm_firstorder_Skewness
5	wavelet-H_glcm_ClusterShade	wavelet-L_glrlm_ShortRunLowGrayLevelEmphasis
6	original_glrlm_ShortRunLowGrayLevelEmphasis	wavelet-H_firstorder_Maximum
7	wavelet-L_glrlm_LongRunLowGrayLevelEmphasis	log-sigma-2-0-mm_firstorder_Mean
8	log-sigma-5-0-mm_firstorder_Skewness	wavelet-H_firstorder_Median
9	log-sigma-3-0-mm_firstorder_Median	wavelet-H_ngtdm_Contrast
10	log-sigma-2-0-mm_ngtdm_Strength	wavelet-L_glrlm_LongRunLowGrayLevelEmphasis
	fold 5	
1	wavelet-H_glcm_Correlation	
2	original_glrlm_LongRunLowGrayLevelEmphasis	
3	wavelet-H_firstorder_Maximum	
4	wavelet-H_glcm_ClusterShade	
5	original_shape2D_MaximumDiameter	
6	wavelet-L_gldm_LargeDependenceLowGrayLevelEmphasis	
7	log-sigma-5-0-mm_firstorder_Maximum	
8	wavelet-H_glcm_Imc2	
9	log-sigma-2-0-mm_firstorder_Skewness	
10	original_glcm_Idmn	

## Data Availability

The dataset analyzed in this study, OPTIMAM Mammography Image Database (OMI-DB), is available from (https://medphys.royalsurrey.nhs.uk/omidb/) upon reasonable request.

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
