# Peer review of "Predicting Breast Cancer Risk Using Radiomics Features of Mammography Images"

_jpm, 2023, doi:10.3390/jpm13111528_

Round 1

Reviewer 1 Report

Comments and Suggestions for Authors

This study is an intriguing investigation with the aim of establishing a method that utilizes radiomic features from mammographic images to predict the risk of breast cancer. The objective is to conduct further examinations and preventive treatments to reduce breast cancer mortality. This implies that a single radiomic feature may serve as an indicator of breast cancer risk. Additionally, the odds ratio for breast cancer incidence was 7.38 among women whose estimated malignancy probability was ≥0.95. Radiomic features derived from mammography images can prove valuable in predicting the risk of breast cancer development. 

Please ask the author to proofread the text for typos, such as repeated keywords and unnecessary numbering.

Author Response

I would like to express my appreciation to the reviewers for their invaluable comments. The precious comments helped us greatly improve our manuscript.

[Comments and Suggestions for Authors]

This study is an intriguing investigation with the aim of establishing a method that utilizes radiomic features from mammographic images to predict the risk of breast cancer. The objective is to conduct further examinations and preventive treatments to reduce breast cancer mortality. This implies that a single radiomic feature may serve as an indicator of breast cancer risk. Additionally, the odds ratio for breast cancer incidence was 7.38 among women whose estimated malignancy probability was ≥0.95. Radiomic features derived from mammography images can prove valuable in predicting the risk of breast cancer development.

Please ask the author to proofread the text for typos, such as repeated keywords and unnecessary numbering.

Response: Thank you for pointing this out. We agree with this comment. Therefore, we have removed duplicate "keywords" and numbering and sorted them alphabetically. [Page 1 , Line 31]

Reviewer 2 Report

Comments and Suggestions for Authors

The manuscript entitled “Predicting breast cancer risk using radiomics features of mammography images “ has been investigated in detail. The topic addressed in the manuscript is potentially interesting and the manuscript contains some practical meanings, however, there are some issues which should be addressed by the authors:

1.      Keywords should be written in alphabetical order.

2.      Delete the “keyword” word in Keywords.

3.      The “Analysis” section is very poor. Please explain more.

4.      The linguistic quality needs improvement. It is essential to make sure that the manuscript reads smoothly- this definitely helps the reader fully appreciate your research findings. There are grammar and writing style errors that should be corrected by the authors.

5.      Some paragraphs are too long to read. They should be divided into two or more for comprehensibility and readability.

6.      How do authors consider the parameters of optimized methods? The values of parameters were selected arbitrarily. The authors provide no guidance or references on how and why to select these values.

7.      What are the other possible methodologies that can be used to achieve your objective in relation in this work?

8.      How the best structure for LGBM  model has been obtained?

9.      Why “k” was considered 5 in this study? Explain in the text.

10.  A deep neural network can have very better results in this case. Why we need decision tree?

Comments on the Quality of English Language

The linguistic quality needs improvement. It is essential to make sure that the manuscript reads smoothly- this definitely helps the reader fully appreciate your research findings. There are grammar and writing style errors that should be corrected by the authors

Author Response

I would like to express my appreciation to the reviewers for their invaluable comments. The precious comments have guided our manuscript toward improvement, making it more informative.

[Comments and Suggestions for Authors]

The manuscript entitled “Predicting breast cancer risk using radiomics features of mammography images “ has been investigated in detail. The topic addressed in the manuscript is potentially interesting and the manuscript contains some practical meanings, however, there are some issues which should be addressed by the authors:

[Comment 1] Keywords should be written in alphabetical order.

Response: Thank you for pointing this out. We have sorted the keywords alphabetically.

[Comment 2] Delete the “keyword” word in Keywords.

Response: Thank you for pointing this out. We have removed the duplicate "keywords".

[Comment 3] The “Analysis” section is very poor. Please explain more.

Response: We have made extensive additions to the Analysis section. We added a one-sample t-test for the AUC for each age group and an age-adjusted analysis of covariance for the difference between the normal and malignant groups for each radiomics feature.

We revised the text as follows:

“All machine learning analyses and statistical processing were performed in Python 3.6.5.

We used the Light Gradient Boosting Machine (LGBM) (https://lightgbm.readthedocs.io/en/v3.3.2/) for our classification task. LGBM is an open-source distributed gradient-boosting framework developed by Microsoft Corporation that uses supervised learning to compute the objective variable from the explanatory variables using a decision tree method. Unlike the sort-based decision tree algorithm used in eXtreme Gradient Boosting and other implementations, LGBM features a highly optimized histogram-based decision tree learning algorithm, which increases efficiency and reduces memory consumption.

We conducted the performance evaluation of the classification task using a k-fold cross-validation scheme. The cases were randomly divided into k equal subsets, and the evaluation was performed k times, each time using a different subset as the test set. In this study, k was set to 5 to ensure a sufficient number of normal and malignant cases in each subset. The k-fold cross-validation scheme is effective in preventing models from overfitting.

We trained and tested the LGBM with the target variable being whether the case was malignant or normal, and the explanatory variable being radiomics features. We calculated the accuracy and AUC for each split. Due to the age bias in the dataset, we conducted analyses limited to all ages and individuals in their 50s or 60s to control for age effects. We verified whether the AUC for each age group significantly exceeded 0.5 using a one-sample t-test.

The probability of malignancy for each case was predicted using the trained LGBM. We predicted the probability of each case being a malignant case using a trained LGBM. The threshold for the malignant probability for calculating the odds ratio of being the contralateral breast of a malignant tumor was set at 0.95.

In the classification task distinguishing malignant from normal cases, we identified the radiomics features that were frequently involved in the decision tree branching. To verify the distribution differences of these radiomics features between malignant and normal cases, we conducted a covariance analysis adjusted for age.” [Page 5, Line 174-203]

[Comment 4] The linguistic quality needs improvement. It is essential to make sure that the manuscript reads smoothly- this definitely helps the reader fully appreciate your research findings. There are grammar and writing style errors that should be corrected by the authors.

Response: Thank you for pointing this out. We asked an English editing service to re-edit our manuscript.

[Comment 5] Some paragraphs are too long to read. They should be divided into two or more for comprehensibility and readability.

Response: Thank you for pointing this out. We agree with this comment. We have revised the Analysis section, as well as divided the paragraphs appropriately [Page 5, Line 174-203]. We also used an editing service to improve the readability of each paragraph.

[Comment 6] How do authors consider the parameters of optimized methods? The values of parameters were selected arbitrarily. The authors provide no guidance or references on how and why to select these values.

Response: Thank you for pointing this out. We primarily utilized the default parameters for Light GBM, with a few specific modifications. We set random_state to 0 to ensure the reproducibility of the experiment, and class_weight was set to “balanced” to account for the disparity in the number of normal and malignant cases. The learning_rate was set to 0.0001, instead of the default value of 0.1, to suppress significant instability among folds with the default setting. Given the low learning rate, we set n_estimators to a large value of 1,000,000, with an early stopping.

[Comment 7] What are the other possible methodologies that can be used to achieve your objective in relation in this work?

Response: Thank you for your question. We believe, as you pointed out in comment 10, that breast cancer risk could still be predicted from mammography images with deep learning. However, we believe that our current approach is sufficiently concise to achieve our objective. Application of methodologies other than radiomics will be our future work.

[Comment 8] How the best structure for LGBM model has been obtained?

Response: Thank you for your question. This time we have not optimized the LGBM structure nor its parameters but used it as-is. Please see our response to "Comment 6" for more information on setting parameters.

[Comment 9] Why “k” was considered 5 in this study? Explain in the text.

Response: Thank you for your question. We revised the text as follows:

"We conducted the performance evaluation of the classification task using a k-fold cross-validation scheme. The cases were randomly divided into k equal subsets, and the evaluation was performed k times, each time using a different subset as the test set. In this study, k was set to 5 to ensure a sufficient number of normal and malignant cases in each subset. The k-fold cross-validation scheme is effective in preventing models from overfitting." [Page 5, Line 184-189]

[Comment 10] A deep neural network can have very better results in this case. Why we need decision tree?

Response: We concur with your comment that deep learning could potentially predict breast cancer risk from mammography images. However, there is a reason for using the decision tree method in this study. We have added an explanation in the text as follows:

"Deep learning-based methods might also be able to predict the risk of developing breast cancer using mammography images of the contralateral breast. Nevertheless, for the classification tasks in this study, we chose to use LGBM, a decision tree method. There are two reasons for this. One is that there have been reports that machine learning is more useful than deep learning when the difference between the number of radiomics features used as explanatory variables and the number of cases is not sufficient [45]. In this study, we used a large number of radiomics features, so we decided to use the decision tree method. Another reason is that using decision tree methods allows us to provide explanations for classification using radiomics features. It is challenging to explain which elements of each image were used for classification in the deep learning method. In fact, we were able to identify several radiomics features, such as ‘wavelet-H_glcm_Correlation’ and ‘wavelet-H_firstorder_Maximum’, which may be useful in predicting the risk of breast cancer development. It is desirable to verify whether these features are related to known breast cancer risk factors, such as genetic mutations [16], smoking [41], and long-term estrogen exposure [39], in the future."

[Page 10, Line 250-265]

[Comments on the Quality of English Language]

The linguistic quality needs improvement. It is essential to make sure that the manuscript reads smoothly- this definitely helps the reader fully appreciate your research findings. There are grammar and writing style errors that should be corrected by the authors.

Response: Thank you for pointing this out. We asked an English editing service to re-edit our manuscript. I think the English in our manuscript has now considerably improved.

Reviewer 3 Report

Comments and Suggestions for Authors

The manuscript "Predicting breast cancer risk using radiomics features of mammography images” aims to establish a method for predicting the risk of developing breast cancer based on radiomics features extracted from mammography images. The study's objective is relevant and of substantial importance in the field of cancer research. As breast cancer represent an important health concern, the potential to predict its risk through non-invasive methods like radiomics features holds great promise. I appreciate the authors' efforts to address this important healthcare challenge that could potentially contribute to the early detection and reduction of breast cancer mortality.

I consider that the work has been well-conceived and methodologically adequately addressed. My only concern is the quality of the English language, which does not meet the standards expected by the journal. I believe that the manuscript should undergo a thorough proofreading and editing by a native speaker, and that, with minor revisions, it has the potential to be published.

Furthermore, I would like to offer two suggestions for the author's future work:

Model Performance: The average Area Under the Curve (AUC) of 0.60122, though statistically significant, indicates only moderate predictive performance. Further improvements in model accuracy would strengthen the study's potential clinical impact.

External Validation: The study relies on a single dataset, the OPTIMAM dataset, which may have inherent biases. The findings should be validated on independent datasets to ensure their generalizability.

Comments on the Quality of English Language

Manuscript should undergo a thorough proofreading and editing by a native speaker.

Author Response

I would like to express my appreciation to the reviewers for their invaluable comments. The precious comments have guided our manuscript toward improvement, making it more informative.

[Comments and Suggestions for Authors]

The manuscript "Predicting breast cancer risk using radiomics features of mammography images” aims to establish a method for predicting the risk of developing breast cancer based on radiomics features extracted from mammography images. The study's objective is relevant and of substantial importance in the field of cancer research. As breast cancer represent an important health concern, the potential to predict its risk through non-invasive methods like radiomics features holds great promise. I appreciate the authors' efforts to address this important healthcare challenge that could potentially contribute to the early detection and reduction of breast cancer mortality.

I consider that the work has been well-conceived and methodologically adequately addressed. My only concern is the quality of the English language, which does not meet the standards expected by the journal. I believe that the manuscript should undergo a thorough proofreading and editing by a native speaker, and that, with minor revisions, it has the potential to be published.

Furthermore, I would like to offer two suggestions for the author's future work:

Model Performance: The average Area Under the Curve (AUC) of 0.60122, though statistically significant, indicates only moderate predictive performance. Further improvements in model accuracy would strengthen the study's potential clinical impact.

External Validation: The study relies on a single dataset, the OPTIMAM dataset, which may have inherent biases. The findings should be validated on independent datasets to ensure their generalizability.

Response: Thank you for your valuable comments. We asked an English editing service to re-edit our manuscript. I think the English in our manuscript has since improved.

I greatly appreciate your advice regarding future work. We agree with it. We would like to further improve the accuracy and consider validation on other datasets.

Round 2

Reviewer 2 Report

Comments and Suggestions for Authors

My recommendation is "Accept in present Form".

Author Response

I would like to express my appreciation to the reviewers for their invaluable comments. The precious comments helped us greatly improve our manuscript.

[Comments and Suggestions for Authors]

My recommendation is "Accept in present Form".

Response: Your valuable comments have guided our manuscript toward improvement, making it more informative.